# Motivation and Limiting Factors for Adherence to Weight Loss Interventions among Patients with Obesity in Primary Care

**DOI:** 10.3390/nu14142928

**Published:** 2022-07-17

**Authors:** Nuria Trujillo-Garrido, María J. Santi-Cano

**Affiliations:** 1Nursing and Physiotherapy Department, School of Nursing, University of Cádiz, C/Venus s/n, 11207 Cádiz, Spain; 2Institute of Biomedical Research and Innovation of Cádiz (INiBICA), Puerto del Mar Hospital, Avda. Ana de Viya 21, 11009 Cádiz, Spain; mariajose.santi@uca.es; 3Research Group on Nutrition: Molecular, Pathophysiological and Social Issues, University of Cádiz, Avda. Ana de Viya 52, 11009 Cádiz, Spain; 4Nursing and Physiotherapy Department, School of Nursing and Physiotherapy, University of Cádiz, Avda. Ana de Viya 52, 11009 Cádiz, Spain

**Keywords:** obesity, motivation, limiting factors, weight management

## Abstract

The cornerstones of obesity management are diet, physical activity and behavioral change. However, there is considerable scientific evidence that lifestyle interventions to treat obesity are rarely implemented in primary care. The aim of this study is to analyze motivation to lose weight among patients with obesity, the resources implemented by primary care centers to promote behavioral change and the limiting factors reported by the patients themselves when attempting to lose weight. A total of 209 patients diagnosed with obesity were interviewed. The variables were obtained from both electronic clinical records (sex, age, BMI, diagnosis of metabolic syndrome and records of activities prescribed to promote behavioral change) and a self-administered personal questionnaire. A total of 67.5% of the respondents reported not having sufficient motivation to adhere to a weight loss program. Records of behavioral change activities were identified in only 3% of the clinical records reviewed. The barriers to adherence to diet and exercise plans most frequently mentioned by patients were not having a prescribed diet (27.8%), joint pain (17.7%), getting tired or bored of dieting (14.8%) and laziness (11.5%). Both the high percentage of patients reporting insufficient motivation to lose weight and the barriers to weight loss identified suggest that patients feel the need to improve their motivation, which should be promoted through primary care.

## 1. Introduction

Obesity has become one of the greatest public health challenges of the 21st century due to both its high prevalence worldwide and the complexity of its adequate management [1,2].

The cornerstones of obesity management are diet, physical activity and behavioral change [3,4,5,6,7]. Behavioral change interventions are of vital importance and are widely recommended by clinical practice guidelines [2,6,7]. However, despite the abundance of resources and scientific evidence advocating a biopsychosocial approach to obesity management, there is also a wealth of evidence suggesting that obesity is not being adequately managed in primary care and that lifestyle interventions to treat obesity are rarely implemented [8,9,10,11]. These limitations in obesity treatment have been and continue to be the subject of research in an attempt to clarify the causes identified by healthcare professionals when treating obesity. Limiting factors mentioned include lack of training (especially in the field of behavioral counseling) and limited consultation time. Another of the barriers most frequently mentioned by healthcare professionals which particularly stands out is the fact that patients lack the necessary motivation to lose weight [12,13].

In relation to this phenomenon, numerous studies have also established that people living with obesity often face persistent stigmatization in different settings such as the workplace, educational and even healthcare environments [14]. In this sense, the World Health Organization stressed that “stigma and discrimination in health takes many forms—the denial of health care and unjust barriers to service provision, inferior quality of care and a lack of respect” [15].

For all of the above reasons, we consider it necessary to give a voice to people with obesity in order to analyze the real motivation that they themselves report, along with the perceived limiting factors and barriers to losing weight in primary care settings.

Therefore, the aim of this study is to analyze motivation to lose weight among patients with obesity, the resources implemented by primary care centers to promote behavioral change and the limiting factors reported by the patients themselves when attempting to lose weight.

## 2. Materials and Methods

The study population consisted of patients diagnosed with obesity who attended consultations at a primary care center in Guadalajara (Spain).

### 2.1. Inclusion and Exclusion Criteria

The inclusion criteria were as follows: adults over 18 years old diagnosed with obesity (defined as BMI ≥ 30 kg/m^2^) over a period of at least five years prior to the study. The exclusion criteria were: obesity secondary to genetic syndromes; hypothalamic or hormonal alterations; any liver, heart or kidney disease causing edema, which could affect body weight and/or waist circumference; terminal illness; cognitive impairment preventing collection of information; pregnancy or breastfeeding; patient absenteeism from primary care consultations for over a year; patients attending day centers for meals or institutionalized elderly people.

### 2.2. Sample Size Determination

Taking into account a population of 510 patients with obesity in 2016 and assuming an estimated proportion of 20% obesity management [16], the sample size required to achieve precision of 4% and a 90% confidence level was 177 patients. We contacted all the eligible participants, and 209 subjects who met the inclusion criteria chose to participate.

### 2.3. Research Variables

Once the patients had given their written consent, data were retrieved from their electronic clinical records: age, sex, BMI and types of behavioral change activities prescribed. Subsequently, further variables were obtained from the personal interview: patient motivation to lose weight, limiting factors identified by the patients and following a diet on their own initiative. An ad hoc survey with one open-ended and five closed-ended questions was administered to the participants and completed in the presence of the principal investigator (Appendix A). To evaluate the responses to the open-ended questions, content analysis of the interview was used to generate a thematic description of the responses given. The responses were recorded verbatim as expressed by the participants.

The content analysis was applied according to the method described by Krippendorf [17]. The unit of analysis was the phrase used by the patients in response to the corresponding question. The common ideas were grouped into categories and subcategories. If more than one category or subcategory was identified in the same response, each of them was classified in its corresponding group.

### 2.4. Ethical Considerations

The study was carried out in compliance with the criteria established by the Declaration of Helsinki (Fortaleza, Brazil) and was approved by the ethics and research committee of the reference hospital (Hospital de Guadalajara). All subjects signed a consent form to participate in the study.

### 2.5. Data Processing and Statistical Analysis

The SPSS v 23.0 program (IBM, Armonk, NY, USA) was used for the statistical analysis. The Student’s *t*-test was used to compare means and the Chi-square test was used to compare percentages. The Mann–Whitney test was used to compare medians when data was not normally distributed.

## 3. Results

### 3.1. General Characteristics of the Participants

The average age of the sample was 66 years. The median BMI was 32.9 kg/m^2^ (Interquartile range = 31.1–36.4), and 90.4% of the participants had been diagnosed with metabolic syndrome. There were records of behavioral change activities in only 3% of the clinical records reviewed (Table 1).

### 3.2. Motivation for Adherence to Weight Loss Interventions

In total, 67.5% of the patients reported not having sufficient motivation to adhere to a weight loss program and 20.5% did not believe they needed to lose weight. Furthermore, 22.4% mentioned having lost weight at some point since the obesity diagnosis by following a diet on their own initiative (Table 2).

### 3.3. Limiting Factors for Weight Loss Mentioned by the Patients

Following analysis of the responses to the open-ended question, the limiting factors for weight loss mentioned by the patients were grouped into four categories and 17 subcategories (Table 3).

The reason for not adhering to diets most frequently mentioned by patients was that they had not been provided with a diet at the primary care center (27.8%), followed by difficulties due to getting tired or bored of the diet (14.8%) (Figure 1). The most frequently mentioned perceived barrier to physical exercise was having joint pain (17.7%), and the most frequent barrier with respect to moods was “mood disturbance” (8.1%) (Figure 1). The limiting factors most frequently mentioned by patients in all categories were not having a prescribed diet (27.8%), having joint pain (17.7%), getting tired or bored of dieting (14.8%) and laziness (11.5%) (Figure 1).

### 3.4. Relationship between Limiting Factors and Motivation to Lose Weight, with Sex, Age and BMI of the Participants

With regard to limiting factors for weight loss mentioned by the patients, women had more difficulties related to mood disturbances. Likewise, the participants who mentioned difficulties dieting, exercising and having relative to mood disturbances, had obesity class II (Table 4).

### 3.5. Relationship between Limiting Factors and Motivation to Lose Weight

A total of 44% of the patients that reported not having enough motivation to adhere to a weight-loss plan mentioned having difficulties exercising, and 23.4% mention having difficulties dieting (Table 5).

## 4. Discussion

This study shows poor levels of self-reported motivation to lose weight among patients diagnosed with obesity. They also do not seem to have the necessary support from healthcare professionals in light of the low number of behavioral change interventions prescribed in the clinical records and the fact that the most frequently mentioned barrier to weight loss was “not having a prescribed diet” despite the results of another study published elsewhere using the same population, showing that 79.9% of patients had a diet prescription recorded in their clinical history, and 88.5% had a physical activity prescription [9].

A total of 20.5% of the patients interviewed did not believe that they needed to lose weight, which is particularly striking considering that all have been diagnosed with obesity and 90% had metabolic syndrome at the time of the study (Table 1). Furthermore, 67.5% reported not having sufficient motivation to adhere to a weight loss program. Moreover, older participants tended to have less motivation, finding statistically significant differences (Table 4).

These results are in line with the study by Dicker et al. [18] in which a survey was carried out of people with obesity to identify perceived barriers to weight management, finding that at the time of the interview half of the respondents reported they were not motivated [18]. In the above study, the variables associated with the likelihood of patients being motivated to lose weight fell broadly into three key themes: self-efficacy, setting specific weight loss goals and having a positive and trusting relationship with healthcare professionals [18]. The latter is of the utmost importance, given the social stigma that people with obesity often suffer. In fact, in a UK survey of people with obesity [13], 17% of the participants reported that after discussing the need to lose weight with a healthcare professional they experienced emotions such as embarrassment, and only 36% and 23% of the patients interviewed felt supported and motivated, respectively. Given these results, Hughes et al. concluded that a paradigm shift is needed in the approach to obesity management to avoid the assumption that people with obesity have no interest or motivation to lose weight [13].

The relationship between obesity, social stigma and low self-esteem has been widely studied, along with the impact this has on an individual’s mental health [19,20,21], leading to maladaptive responses such as reduced physical activity and reduced healthcare seeking behavior [14,15].

This decrease in healthcare seeking, which could translate into a low level of self-care, could explain the low percentage of patients who believed they needed to lose weight, as well as the high number of participants who admitted they were not sufficiently motivated to adhere to a weight loss plan.

However, despite the fact that the clinical practice guidelines for the treatment of obesity insist on the need to incorporate psychoeducational interventions to promote adherence to diet and exercise, confidence and intrinsic motivation [2,6,22,23], in the clinical records reviewed examples of such interventions were identified in only 2.9% of cases. Moreover, the interventions recorded consisted of non-specific advice such as chewing slowly or simply changing habits without specifying how. These results are similar to those observed by Sharma et al. [24] in their study regarding perceptions of barriers to effective obesity management, which found that only a minority of the patients and healthcare professionals surveyed (5% and 17%, respectively) admitted to having discussed the possibility of behavior therapy or psychotherapy.

The scientific literature clearly identifies a number of resources in both clinical practice guidelines for obesity management and research articles, such as psychological interventions based on cognitive or problem-solving therapy [2,6,22,23,25,26] and other resources such as mobile applications, group treatment sessions and telephone follow-up to extend the reach of interventions to people in rural areas and reduce costs [27,28,29]. However, the nature of many of these resources raises the need for a multidisciplinary team for obesity management. Indeed, in the study mentioned above by Sharma et al. [24] most of the healthcare professionals mentioned the need for a system for referral to a psychologist or social worker, along with the need for educational programs to better understand the pathophysiology of obesity [24].

Likewise, with regard to the limiting factors for weight loss mentioned by patients in the personal interview, the responses categorized as 1.1. “Getting tired or bored of the diet” (14.8%), 4.1. “Needing more support or external control to change habits” (7.6%) and 3.2. “Feeling too old to diet or do exercise” (2.4%) could be explained by this apparent lack of psychosocial interventions in obesity management, again highlighting the need for more people-focused interventions. It could also explain why 22.4% of the patients have at some point resorted to dieting on their own initiative without the intervention of their primary care physician or endocrinologist (Table 2), which could indicate that many patients prefer to seek support outside the National Health System, since health care by nutritionists or dieticians is not available in the Spanish health system. Another possible reason for resorting to such weight loss plans could be the fact that intensive follow-up through motivational interviewing is of vital importance for effective and sustainable weight loss programs [30,31]. However, it has been found that this does not happen very often [9,24].

As regards the reasons for not doing physical exercise reported by patients, the most frequently given reason was 2.1. “Having joint pains”. It should be remembered that the average age of the study population is M = 65.7, SD = 12.7 years and that arthropathy is a frequent comorbidity of obesity [28,32]. The response categorized as 2.5. “Having an illness that prevents or hinders physical exercise” (2.4%) could also be related to the advanced average age of the sample.

The responses categorized as 2.2 “Laziness” (11.5%), 2.3. “Lack of time” (8.1%) and 2.4. “Exercising only in good weather” once again point to the existence of a lack of motivation and external support as in the case of dieting. These results are in line with the study by Lim et al. [21], which found that time and relational barriers were among the perceived limiting factors to obesity management. This same study also mentions other limiting factors that are consistent with those identified in our study, such as prioritizing children’s food preferences, financial cost and depressive and defeating thoughts related, respectively, to the subcategories 1.3. “Not being able to cook a different menu for yourself” (4.8%), 1.4. “Lack of income” (2.4%), 3.1. “Suffering from mood disturbance” (8.1%) and 3.2. “Feeling very old” (2.4%). These limiting factors identified in our study highlight the limited knowledge the study population has regarding nutrition and healthy eating. This is also particularly striking given that our study has identified other misconceptions or false beliefs, such as the responses categorized as 1.2. “Not wanting to give up eating what you want” (4.8%), subcategory 1.3. “Not being able to cook a different menu for yourself” (4.8%), subcategory 1.4. “Lack of income” (2.4%) and subcategory 1.5. “Not wanting to go hungry” (2%). These assumptions suggest that there is a need to further develop the concepts of healthy diet and healthy living. Response 1.2. in particular, “Not wanting to give up eating what you want” (4.8%), is in line with the study by Borazjani et al. [33] which found that the restrictions imposed by the diets prescribed and the fact that patients’ tastes and preferences were not taken into account in their preparation constituted a major limiting factor for patients.

Special mention should also be made of the response “Lack of income”. Patients who gave this reason for not adhering to a diet explained that in order to lose weight they had to eat mostly protein foods and few carbohydrates and that with their income they could not afford to eat meat and/or fish on a daily basis, which may be a misconception deriving from the popular ketogenic and high-protein diets [34]. Borazjani et al. [33] also identified financial barriers to adhering to a weight loss plan, with some participants in their study mentioning that they could not afford to buy the food that the diets recommended.

The response categorized as 1.6. “Resisting going on a diet until you get sick” could be explained by the fact that according to Kalra et al., the comorbidities associated with obesity, as well as the discomfort and disability associated with them, can also act as motivational cues [35].

Finally, although the study population has an average age of 65 years, the results obtained are similar to those of other studies [18,21,24,35], so the results obtained could be considered generalizable.

## 5. Conclusions

Both the high percentage of patients reporting insufficient motivation to lose weight and the barriers to weight loss identified suggest that patients feel the need to improve their motivation. However, this need does not seem to be supported by motivational interventions in their clinical records. We therefore consider it vitally important to implement biopsychosocial interventions for obesity management through interdisciplinary teams to address behavioral change and motivational interviewing, and, also, to address the possible effects that the stigma of obesity may have on the mental health of patients.

## Figures and Tables

**Figure 1 nutrients-14-02928-f001:**
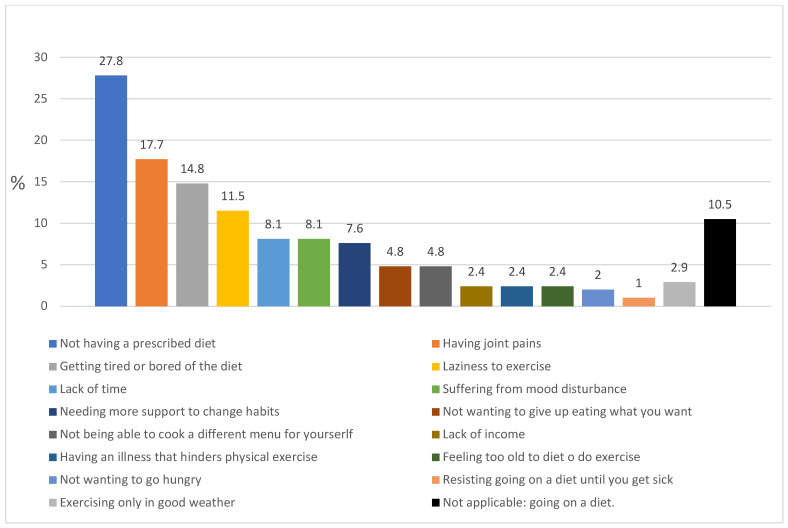
Limiting factors mentioned by patients.

**Table 1 nutrients-14-02928-t001:** General characteristics of the participants.

Characteristics		Total *n* = 209	Men *n* = 88 (42.1%)	Women *n* = 121 (57.9%)	*p*
Age (years) Mean (SD)		65.7 (12.7)	63.2 (13.6)	67.6 (11.7)	0.015
BMI (kg/m^2^) ^a^ Median (Interquartile range)		32.9 (31.1–36.4)	32.2 (30.9–35.0)	33.4 (31.4–36.9)	0.178
Patients’ distribution by BMI classes ^b^ % (*n*)	Overweight	12.4 (26)	15.9 (14)	9.9 (12)	0.178
	Obesity I	53.6 (112)	58.0 (51)	50.4 (61)	
	Obesity II	22.5 (47)	18.2 (16)	25.6 (31)	
	Obesity III	11.5 (24)	8.0 (7)	14.0 (17)	
Metabolic syndrome ^b^ % (*n*)		90.4 (189)	94.3 (83)	87.6 (106)	0.085
Metabolic syndrome diagnosis criteria	Waist circumference Mean (SD)	111.4 (11.6)	116.2 (10.1)	107.9 (11.5)	<0.0001
	High blood pressure % (*n*)	85.6 (179)	89.8 (79)	82.6 (100)	0.166
	Diabetes Mellitus II % (*n*)	38.8 (81)	38.6 (34)	38.8 (47)	1.000
	Hypertrigliceridemia % (*n*)	31.7 (66)	41.4 (36)	24.8 (30)	0.009
	Reduced HDL % (*n*)	53.5 (112)	52.8 (46)	52.5 (62)	1.000
Prescription of behavior change recorded % (*n*)		2.9 (6)	3.4 (3)	2.4 (3)	0.698
Type of behavior change recorded % (*n*)		1.5 (2)	1.8 (1)	1.3 (1)	1.000

SD, standard deviation; BMI, body mass index; Student’s *t*-test.; ^a^ Mann–Whitney U test; ^b^ Chi-squared test.

**Table 2 nutrients-14-02928-t002:** Motivation and factors related to adherence to a weight loss plan.

	Total (*n* = 209)	Men (*n* = 88)	Women (*n* = 121)	*p*
Not having sufficient motivation to adhere to a weight loss program, % (*n*)	67.5 (141)	61.4 (54)	71.9 (87)	0.135
Not believe they need to lose weight, % (*n*)	20.5 (43)	20.5 (18)	20.7 (25)	0.154
Report following a diet on their own initiative, % (*n*)	22.4 (47)	19.3 (17)	24.7 (30)	0.403

Chi-squared test.

**Table 3 nutrients-14-02928-t003:** Limiting factors for weight loss mentioned by the patients. Categories and subcategories.

1. DIFFICULTIES DIETING.
1.1. Getting tired or bored of the diet. 1.2. Not wanting to give up eating what you want. 1.3. Not being able to cook a different menu for yourself. 1.4. Lack of income. 1.5. Not wanting to go hungry. 1.6. Resisting going on a diet until you get sick. 1.7. Not been provided with a diet at the primary care center.
2. DIFFICULTIES EXERCISING.
2.1. Having joint pains. 2.2. Laziness. 2.3. Lack of time 2.4. Exercising only in good weather. 2.5. Having an illness that prevents or hinders physical exercise.
3. MOOD THAT MAKES IT DIFFICULT TO FOLLOW A WEIGHT LOSS PLAN.
3.1. Suffering from mood disturbance. 3.2. Feeling too old to diet or do exercise.
4. DIFFICULTIES CHANGING HABITS.
4.1. Needing more support or external control to change habits.

**Table 4 nutrients-14-02928-t004:** Relationship between limiting factors and motivation to lose weight, with sex, age and BMI of the participants.

		Not Having Sufficient Motivation to Adhere to a Weight Loss Program vs. Having Motivation.	*p*	Difficulties to Dieting vs. * not Having.	*p*	Difficulties to Exercise vs. not Having.	*p*	Difficulties Relative to Mood Disturbance vs. not Having.	*p*	Difficulties to Change Habits vs. not Having.	*p*
Sex ^a^ % (*n*)	Men	61.4 (54)	0.135	25.0 (22)	0.874	43.2 (38)	0.887	1.1 (1)	0.001	6.8 (6)	0.617
	Women	71.9 (87)		26.4 (32)		41.3 50		14.0 (17)		9.1 11	
Age (years) ^b^ Mean (SD)		67.2.7 vs. 62.7 (11.7)/(14.2)	0.015	62.5 vs. 66.9 (12.7)/(12.5)	0.027	62.6 vs. 68.0 (13.1)/(12.0)	0.002	67.8 vs. 65.5 (13.5)/(12.6)	0.466	62.7 vs. 66.0 (14.4)/(12.5)	0.299
BMI (kg/m^2^) ^c^ Median (Interquartile range)		32.4 vs. 33.26 (31.0–36.2)/(31.1–37.1)	0.550	34.9 vs. 32.2 (32.1–37.8)/(30.9–35.4)	0.005	34.4 vs. 31.8 (32.2–37.7)/(30.5–35.0)	0.000	36.4 vs. 32.6 (32.5–39.7)/(31.0–35.7)	0.013	34.4 vs. 32.5 (32.4–35.0)/(31.0–36.5)	0.350

^a^ Chi-squared test; ^b^ Student´s *t*-test; ^c^ Mann–Whitney U test; * Versus.

**Table 5 nutrients-14-02928-t005:** Relationship between limiting factors and motivation to lose weight.

	Not Having Sufficient Motivation to Adhere to a Weight Loss Program vs. * Having motivation.	*p*
Difficulties dieting % (*n*)	23.4 vs. 30.9 (33)/(21)	0.311
Difficulties exercising % (*n*)	44.0 vs. 38.2 (62)/(26)	0.458
Mood % (*n*)	11.3 vs. 2.9 (16)/(2)	0.062
Difficulties changing habits % (*n*)	9.2 vs. 5.9 (13)/(4)	0.590

Chi-squared test; * Versus.

## Data Availability

Due to the sensitive nature of the questions asked in this study, survey respondents were assured raw data would remain confidential and would not be shared.

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
