# Peer review of "Motivation and Limiting Factors for Adherence to Weight Loss Interventions among Patients with Obesity in Primary Care"

_nutrients, 2022, doi:10.3390/nu14142928_

Round 1

Reviewer 1 Report

Reviewer

In this study, the authors analyzed the motivation to lose weight among patients with obesity, the resources implemented by primary care centers to promote behavioral change and the limiting factors reported by the patients themselves when attempting to lose weight. 

Minor revisions need to be done before acceptance.

1)  General Revision:

Typography: the authors should read thoroughly their manuscript and check: 1) space between words; 2) English of some sentences

2)  Material and Methods section:

- We suggest to divide the section into subparagraphs for a better understanding of the text (such as “Data processing and statistical analyisis”)

Reviewer 2 Report

This paper provides insight into the motivations (or lack of motivations) of a primary-care population in Spain. My main concerns are:

1) How generalizable are the results? The population is older than the patients that I see. Most had metabolic syndrome; how bad was it? How many subjects had diabetes?

2) What sort of advice had been given to the patients previously? How many had been prescribed diets or exercise regimens? This may well color their perceptions and willingness to repeat those interventions.

Reviewer 3 Report

The manuscript presents relevant but insufficiently explored data. In its present version it adds little to the body of knowledge, but its implications to practice would greatly improve if the authors included (a) the relationships of motivation and limiting factors with sex, age and BMI; and (b) the relationships between motivation and limiting factors.

The remaining sections must consider these new results.

Besides this major issue, some others should also be addressed:

1. Considering the description, 510 patients do not seem to be a cohort, so please replace that designation in line 70.

2. Was normality assessed? I f yes, please report how. Not normal data should not be described as means and SD nor compared using t tests.

3. Please describe patients' distribution by BMI classes.

4. In table 1 add "%" to values 42.1 and 57.9 (proportion of males and females).

5. In tables 1 and 2 (and in every additional analysis) report the exact values of p (not only "ns").

6. In table 1, line 102, translate into English "T de Student. * Chi cuadrado", and in appendix A delete the "?" before "Have you lose weight...".

7. Figure 1 could be replaced by one single chart, with all limiting factors.

8. Lines 141-143: the justification cannot be because mean BMI belongs to class 1 obesity, as this is the lowest BMI obesity class. Please refer to the fact that all have been diagnosed with obesity and 90% had MS.

9. As the paper will be read by people from different countries, please include some description regarding the availability if nutritionists/ dieticians in primary care in Spain (in lines 193-4 the authors refer to diets prescribed by physicians or endocrinologists, but not to nutritionists/ dieticians).

Round 2

Reviewer 2 Report

Thanks for providing the requested information. Good luck with your future studies in this area.

Reviewer 3 Report

1. Please uniformize the corrections regarding normality. In table 1 BMI is described as median (IQR), but in the text it remains as mean (SD). MW's test should be included in the statistics section.

2. Please replace "gender" (it appears 3 times in current version) with "sex".
